# Microbial Communities of Artisanal Fermented Milk Products from Russia

**DOI:** 10.3390/microorganisms10112140

**Published:** 2022-10-29

**Authors:** Tatiana V. Kochetkova, Ilya P. Grabarnik, Alexandra A. Klyukina, Kseniya S. Zayulina, Ivan M. Elizarov, Oksana O. Shestakova, Liliya A. Gavirova, Anastasia D. Malysheva, Polina A. Shcherbakova, Darima D. Barkhutova, Olga V. Karnachuk, Andrey I. Shestakov, Alexander G. Elcheninov, Ilya V. Kublanov

**Affiliations:** 1Winogradsky Institute of Microbiology, Federal Research Center of Biotechnology of the Russian Academy of Sciences, 117312 Moscow, Russia; 2Applied Genomics Laboratory, SCAMT Institute, ITMO University, 197101 Saint Petersburg, Russia; 3Faculty of Biology, Lomonosov Moscow State University, 119234 Moscow, Russia; 4Institute of General and Experimental Biology Siberian Branch of the Russian Academy of Sciences, 600047 Ulan-Ude, Russia; 5Laboratory of Biochemistry and Molecular Biology, Tomsk State University, 634050 Tomsk, Russia

**Keywords:** fermented milk products, NGS, fermented beverages, traditional dairy products, microbial community, lactic acid bacteria

## Abstract

Fermented milk products (FMPs) have numerous health properties, making them an important part of our nutrient budget. Based on traditions, history and geography, there are different preferences and recipes for FMP preparation in distinct regions of the world and Russia in particular. A number of dairy products, both widely occurring and region-specific, were sampled in the households and local markets of the Caucasus republics, Buryatia, Altai, and the Far East and European regions of Russia. The examined FMPs were produced from cow, camel, mare’s or mixed milk, in the traditional way, without adding commercial starter cultures. Lactate and acetate were the major volatile fatty acids (VFA) of the studied FMPs, while succinate, formate, propionate and *n*-butyrate were present in lower concentrations. Bacterial communities analyzed by 16S rRNA gene V4 fragment amplicon sequencing showed that *Firmicutes* (*Lactococcus*, *Lactobacillus, Streptococcus*, *Lentilactobacillus* and *Leuconostoc*) was the predominant phylum in all analyzed FMPs, followed by *Proteobacteria* (*Acetobacter, Klebsiella, Pseudomonas* and *Citrobacter*). *Lactobacillus* (mainly in beverages) or *Lactococcus* (mainly in creamy and solid products) were the most abundant community-forming genera in FMPs where raw milk was used and fermentation took place at (or below) room temperature. In turn, representatives of *Streptococcus* genus dominated the FMPs made from melted or pasteurized milk and fermented at elevated temperatures (such as ryazhenka, cottage cheese and matsoni-like products). It was revealed that the microbial diversity of koumiss, shubat, ryazhenka, matsoni-like products, chegen, sour cream and bryndza varied slightly within each type and correlated well with the same products from other regions and countries. On the other hand, the microbiomes of kefir, prostokvasha, ayran, cottage cheese and suluguni-like cheese were more variable and were shaped by the influence of particular factors linked with regional differences and traditions expressed in specificities in the production process. The microbial diversity of aarts, khurunga, khuruud, tan, ayran and suluguni-like cheese was studied here, to our knowledge, for the first time. The results of this study emphasize the overall similarity of the microbial communities of various FMPs on the one hand, and specificities of regional products on the other. The latter are of particular value in the age of globalization when people have begun searching for new and unusual products and properties. Speaking more specifically, these novel products, with their characteristic communities, might be used for the development of novel microbial associations (i.e., starters) to produce novel products with improved or unique properties.

## 1. Introduction

For centuries, animal-based fermented milk products (FMPs) have been consumed by people from various ethnic groups and origins. Currently, traditional FMPs are still a major foodstuff for their high nutritional value and health-promoting properties [1]. The fermentation process increases the shelf-life of these products while also enhancing the taste and improving the digestibility of milk. Home-made products are fermented spontaneously by natural starter cultures, usually by inoculating raw milk with the previous batch’s fermented milk product (back-slopping method) [2]. Over time, starters which reduce the fermentation time and improve the quality and taste of the product have been selected. This has led to sustainable microbial starter communities and consequently FMPs that are well adapted to certain conditions. Cow’s milk and its derivatives (cream, serum, etc.) are most often used to make FMPs, whereas milk from sheep, goats, mares, buffaloes and other animals is less common and basically linked to a specific region. The variety of FMPs is extremely large and depends on microbiome composition, production technology and raw materials, which, in turn, greatly depends on national/regional traditions. Russia consists of more than 190 different ethnic groups, living in 85 regions, which have their own traditions in preparing dairy products, resulting in dozens of different types of FMPs, including the widespread and common kefir, ryazhenka, sour cream, cottage cheese, yogurt, sour milk and various cheeses as well as rare products as katyk, suzma, ayran, tarag, khurunga etc. [3].

Early studies on FMPs microbial communities was started at the end of the 19th century and the very first pure bacterial culture, isolated and described by Joseph Lister, was a lactic acid bacterium, currently known as *Lactococcus lactis* [4]. Stanislav Grigorov discovered that lactic acid bacteria (LAB) are involved in milk fermentation, and further research by Ilya Mechnikov on microbial antagonism and its benefits for human health led to extensive studies of the microbial diversity of FMPs. Since then, using standard microbiological cultivation-based approaches, it has been revealed that representatives of LAB, propionic bacteria and yeasts dominate in different types of FMPs. However, the entire microbial diversity, including taxa that are rare and uncultivated in standard media, could not be assessed using these approaches [5]. In recent decades, with the development of molecular ecological techniques based on DNA sequencing (first of all, next generation sequencing, NGS), it has become possible to study the microbial communities while avoiding cultivation [5,6,7]. Using these approaches, knowledge of the diversity of the microbial communities of FMPs has been greatly improved. The group of lactic acid bacteria, well known to dominate in the majority of FMPs, was expanded to hundreds of species, most of which belonged to the genera *Lactobacillus, Leuconostoc, Pediococcus, Lactococcus, Enterococcus, Weissella,* or *Oenococcus*. Besides LAB, representatives of other bacteria, such as *Bifidobacterium, Acetobacter, Alistipes, Allobacter, Bacteroides,* and *Gluconobacter* were detected to be involved in various types of FMPs from many regions [5,7,8,9,10,11]. NGS helps to control the process of FMP production; the detection of the presence of pathogenic microbiota or ‘unwanted’ microorganisms involved in spoilage has become much easier. For example, the detection of bacteria belonging to *Pseudomonadaceae, Enterobacteriaceae,* or *Comamonadaceae* families or some species of *Acinetobacter, Sphingomonas* and *Staphylococcus*, which serve as indicators of the low hygienic quality of milk and/or a technological process, has become much easier [12,13,14]. All of these culture-independent NGS-based studies resulted in the microbial diversity of various fermented dairy products being deciphered worldwide [2,5,7,11,15,16,17,18]. On the other hand, still little is known about the microbiomes of many local FMPs. This is correct for Russian–made FMPs, the microbial communities of which have been almost completely uninvestigated using NGS. Several studies have been performed with traditional artisanal fermented dairy products from cow and mare milk (sour cream, cottage cheese, cheese and koumiss), produced in the Kalmykia, Buryatia and Tuva regions (for example, [19,20,21]). One investigation was made for a number of commercial FMPs [16]. Yet, all these are sporadic, single-case studies.

In this work, a large-scale study of the microbiomes of more than fifty FMPs of animal origin from different regions of the Russian Federation using NGS was obtained. We analyzed the microbial communities of home-made dairy products, such as fermented beverages (kefir, ryazhenka, prostokvasha, aarts, khurunga, koumiss, shubat, ayran and tan), cream-like and curd-like products (analogue of matsoni, sour cream, chegen and cottage cheese) and some cheeses (khuruud, suluguni-like salt cheese and bryndza) from the Baykal region, Altai, the Far East and European regions of Russia and the North Caucasus republics.

## 2. Materials and Methods

### 2.1. Collection of Dairy Product Samples

Fermented milk products prepared by traditional methods were sampled in local markets in villages and towns of various regions of Russia in 2021 and 2022 during the autumn, spring and summer seasons. For DNA fixation, 2 mL aliquots of dairy products were mixed with 2 mL of fixing buffer (100 mM EDTA, 100 mM Tris-HCl, 150 mM NaCl; pH 8.2) at sampling locations. A 20 mL syringe with its front end cut off was used to sample 2 mL of the cheeses. The samples were then transported to the laboratory at 4 °C. DNA extraction and all other manipulations were carried within 7 days after sampling.

### 2.2. Volatile Fatty Acids Analysis

At the initial stage of sample preparation for creamy products and cheeses, approximately 20 mL of each product were pressed through a sieve with a mesh size 0.5 × 0.8 mm into a sterile 50 mL tubes. After measuring the mass of the products, 20 mL of sterile water was added to each sample followed by rigorous stirring and centrifugation at 130× *g* for 1 min in order to get rid of large insoluble particles. The supernatants were transferred into 50 mL tubes, weighted and diluted with 20 mL of sterile water. In the case of beverages, 20 mL of each product was diluted once with 20 mL of water in a sterile 50 mL tube followed by vigorous stirring. All the samples were then centrifuged at 18,500× *g* for 20 min. A total of 4 mL of each supernatant was collected in a 5 mL tube and centrifuged again for 40 min at 18,500× *g* to remove remaining particles and fat. The transparent liquid interphases were collected by 5 mL syringe and filtered through a 0.22 µm pore size PES membrane syringe filter (Millipore) into a 2 mL tube. At this stage, the pH value of the filtrated sample was measured using pH indicator strips (Macherey-Nagel pH-fix 0.0–6.0). To remove soluble proteins, which might interfere with the detection of volatile fatty acids (VFA) and contaminate the column, ice-cold 100% trifluoroacetic acid was added to each sample in 1:10 (v:v), and the mixture was vortexed and left overnight at 4 °C. The samples were centrifuged for 60 min at 18,000× *g* and 4 °C, upon which 0.1 mL of the supernatant of each sample was transferred into a 1.5 mL glass vial and diluted with 0.9 mL of deionized water. Samples from the Altai Republic (78AR, 75CG, 76CG and 77SB) were not analyzed.

VFA in soluble FMPs were analyzed using a 1260 Infinity II liquid chromatograph (Agilent, Santa Clara, CA, USA) and Aminex HPX-87H column (BioRAD, Hercules, CA, USA) with Micro-Guard Cation H guard column set up in isocratic mode with 5 mM H_2_SO_4_ mobile phase at a flow rate of 0.6 mL/min. The absorbance of the eluent was measured at 210 nm (bandwidth 208–212) using a G7115A DAD WR (Agilent, Santa Clara, CA, USA) ultraviolet detector. The samples’ VFA were identified by comparing retention times of the peaks of the samples and the calibration solutions of VFA. 

### 2.3. DNA Extraction and Amplicon Sequencing

Fixed dairy product samples were centrifuged at 18,000× *g* for 20 min and the pellets were used for DNA extraction, which was performed using DNeasy PowerLyzer Microbial Kit (Qiagen, Hilden, Germany) according to the manufacturer’s instructions, including a bead-beating stage using a FastPrep-24™ 5G grinder (MP Bio, Santa Ana, CA, USA). Amplicon libraries of the V4 region of the 16S rRNA gene were prepared as described previously [22] using a pair of primers 515F [23] (5′-GTGBCAGCMGCCGCGGTAA-3′)-Pro-mod-805R [24] (5′-GGACTACHVGGGTWTCTAAT-3′). The libraries were sequenced using a MiSeq system (Illumina, San Diego, CA, USA). The libraries were prepared and sequenced in two replicates for each sample. All sequencing data were deposited into the NCBI SRA database under BioProject number PRJNA789261 (Appendix A).

### 2.4. Bioinformatics and Statistical Analysis

Adapter trimming and demultiplexing were performed as described earlier [25]. The obtained reads were filtered and processed using dada2 package v.1.14.1 [26] (parameters: truncLen = 220, maxN = 0, maxEE = 2, truncQ = 2) resulting in identification of the amplicon sequence variants (ASV). Taxonomic assignment of ASV was performed using dada2 package v.1.14.1 with native Bayesian classifier [27] and Silva 138.1 database [28]. Biodiversity metrics such as Shannon [29], InvSimpson [30] indexes, and Chao1 richness estimator [31] were calculated using the phyloseq v.1.3 package [32]. Statistical analysis of alpha-diversity metrics was performed using a one-way ANOVA test in R stats package (https://www.r-project.org; accessed on 30 September 2022). Rarefaction curves were inferred using iNEXT package v.2.0.20 (q = 0) [33]. To estimate the dissimilarity of the microbial composition of FMPs (i.e., the beta-diversity), a non-metric multidimensional scaling (NMDS) was performed as an ordination method based on ASV summarized table and the Bray-Curtis dissimilarity indices using phyloseq and vegan v.2.6.-2 packages [34]. Visualization of the results was performed with the ggplot2 package (https://ggplot2.tidyverse.org; accessed on 30 September 2022).

## 3. Results and Discussion

We analyzed 55 samples of 16 types of home-made fermented dairy products from 9 regions of Russia (Figure 1, Table 1).

### 3.1. Volatile Fatty Acids Content

The VFAs revealed in the FMPs were formate, acetate, lactate, propionate, *n*-butyrate and succinate. The major VFAs in all samples were lactate and acetate (Figure 2) with concentrations varying from 3.3 to 813.6 mM and 4.9 to 361.1 mM, respectively. Formate and *n*-butyrate concentrations varied from 0.3 to 12.1 mM and 0.27 to 22.6 mM, respectively; propionate and succinate concentrations varied from 0.2 to 40 mM and 0.63 to 59 mM, respectively, depending on the type of FMPs (Appendix A). 

In beverages, the total VFA concentrations were in the range 26.3 to 484.2 mM with the lowest values in tan (26.3 mM) and the highest values in ryazhenka (484.2 mM, Appendix A) and this matched well with the fairly flat taste of the first and the rich taste of the second. Among the creamy products, the total VFA concentrations were in the range of 103.8 in matsoni-like samples to 1035.8 mM in the cottage cheese. In cheese samples, the total VFA concentrations also varied greatly from 163.9 mM in khuruud to 899 mM in bryndza or 865.8 mM in suluguni-like cheese. In cheeses, the concentration of VFA positively correlated with bacterial diversity (see below).

Besides the total amount of VFA, the FMPs differed from each other in the ratio of particular organic acids, which is the main factor responsible for their differences in taste. Beverages from Buryatia—aarts, khurunga and koumiss—were rich in lactate and acetate, but also succinate and formate were measured in some of the samples and 1.41 mM of *n*-butyrate was measured in one khurunga (18KR) sample. In turn, ayran samples from Karachay-Cherkessia and Stavropol Region contained small amounts of succinate (0.2–1.16 mM), but also a small amount of propionate (0.2–0.78 mM), absent in the Buryatia samples. The VFA content of two kefir samples from Stavropol Region (7KF) and Primorsky Region (25KF) were quite different. The sample 7KF was rich in lactate (167.1 mM); acetate (8.2 mM) was the only other detected VFA. In another kefir (25KF), the lactate/acetate ratio was already 3/1 (86.5 mM and 26.3 mM, respectively) and also succinate (0.98 mM) and propionate (0.2 mM) were also detected. Prostokvasha samples from Arkhangelsk and Tula regions and Dagestan Republic were heterogeneous in VFA content. The major VFAs—lactate and acetate—varied between different samples: lactate from 82.3 (37PS) to 190 mM (66PS) and acetate from 0 (74PS) to 68.1 mM (17PS). Succinate, formate and propionate concentrations fluctuated from 0.59 (16PS) to 12.82 mM (29PS), 0.94 (29 PS) to 2.23 mM (17PS) and 0 (17PS, 69PS) to 11.23 mM (39PS), respectively. Ryazhenka samples from different regions were similar to each other in their abundancy of lactate and acetate (yet their ratio was different) but differed in minor VFA: succinate (0–0.63 mM), formate (0–0.32 mM) and propionate (0–3.22 mM). Tan (22TA) from Primorsky Region had the lowest total VFA concentration and besides lactate and acetate also contained minor amounts of formate and propionate (all four in concentrations of 18.8, 5.0, 1.8 and 0.6 mM, respectively).

Matsoni-like products from Karachay-Cherkessia Republic and Stavropol Region (5MC and 13MC, respectively) were characterized by similar concentrations of lactate, acetate, succinate and propionate. In 13MC, trace amounts of formate (0.55 mM) and *n*-butyrate (0.28 mM) were also detected. The VFAs in samples of all sour cream and cottage cheese samples were lactate, acetate, succinate and propionate. These samples differed in concentrations of minor VFAs: propionate and *n*-butyrate. The cottage cheese from Stavropol Region (15TG) had rather a high amount of propionate (17.52 mM) while the cheese from Buryatia (30TG) contained 47.94 mM of succinate.

As in all the other products, the main VFAs in white cheeses (bryndza, suluguni-like and khuruud) from all regions were lactate and acetate. Except the suluguni-like product from Stavropol Region (6SU) containing 38.79 mM of succinate, the concentration of minor VFAs in all other white cheeses did not exceeded 6 mM, and *n*-butyrate was not detected in any of them.

### 3.2. Biodiversity and Taxonomic Profiling of the Studied Fermented Milk Products

A total of 1,453,735 raw reads with an average length of 250 bp were retrieved after sequencing 55 samples. After filtering, denoising and chimera detection, 1,296,815 reads were retained, representing 367 unique sequences (Appendix A). The obtained ASV were assigned to more than two hundred genera within 32 phyla, but more than 95% of the total number of sequences were affiliated to *Firmicutes* and *Proteobacteria* (Figure 3).

Rarefaction curves were plotted to evaluate the depth of sequencing (Appendix A). To estimate overall diversity in all samples analyzed, the alpha-diversity indexes were calculated for each sample. According to the Shannon Index, suluguni-like cheeses, sour cream samples and aarts possessed the highest biodiversity in comparison with other studied products (Figure 4A). The maximum value of 2.72 was observed for suluguni-like cheese (05SU). Bryndza, cottage cheeses, shubat, koumiss and tan had lower biodiversity with 0.99–1.75 (median—1.38), 0.23–2.27 (median—1.41), 1.01, 1.21 and 1.34, respectively. Low median index values were observed for many beverages (ayran—0.7, kefir—0.56, prostokvasha—0.78, ryazhenka—0.79, khurunga—0.88), creamy products (chegen—0.69, and matsoni-like—0.49) and khuruud (0.57). The analysis of the diversity based on the inverse Simpson index and Chao1 richness estimator revealed similar patterns (Figure 4B,C). The highest Chao1 species richness varied from 3 (05AR) to 80 (25AA), while inverse Simpson values were from 1.02 (74PS) to 9.9 (05SU). It should be noted that the variability of biodiversity within some of the studied products (e.g., prostokvasha, sour cream and cottage cheese) was rather high and the overall diversity indexes cannot be applied for these types of samples. Their differences, most probably, are due to some unmeasured properties of milk (e.g., fat content [35]) or production process.

The comparison of the microbiomes of different FMPs was performed with NMDS using Bray–Curtis dissimilarity indices based on an ASV table. The microbial components of each sample (ASV) determine the spatial distribution of points, i.e., the biodiversity of the products. The majority of the samples formed two distinct clusters (Figure 5). The first one included matsoni (MC), almost all ayrans (excluding 78AR), ryazhenka (RZ), tan (TA), kefir 07KF, cottage cheese 03TG and several prostokvasha samples (66PS, 69PS, 74PS). Bryndza (BZ), khuruud (KU), sour cream (SM + SN), suluguni-like 06SU, other prostokvasha (16PS, 29PS, 49PS), kefir 25KF, cottage cheese (16TG, 20TG) and khurunga 18KR were located within the second cluster. Some of the samples were outside of these two main groups, either completely separately, such as 75CG, 76CG, 77SB, 31KM, 25AA, 26KR, or forming a quite uniform gradient of the measure of composition dissimilarity between these clusters, such as 05TG, 06TG, 07TG, 14TG, 15TG, 17TG, 22SM, 06SN, 03SU, 05SU, 37PS, 07BZ, 78AR. NMDS revealed that *Streptococcus* and *Lactococcus* have the greatest impact on clustering and are the main component of the microbial communities for the first and the second clusters, respectively. The distribution of total VFA concentrations also follows the clustering trend: low to moderate total VFA values were observed for the first cluster while moderate-high concentrations were observed for the second cluster. The concentrations of individual VFAs showed similar tendencies as the total concentrations did (Appendix A).

#### 3.2.1. Fermented Milk Beverages

***Aarts*** is a highly nutritious Buryats drink. As for all other Buryats FMPs, the microbial communities of aarts have never been investigated before. For its preparation, “bozo” (curd mass) is mixed with cold water and melted with the addition of wheat flour. *Lactobacillus* (48%), *Acetobacter* (26.9%) and *Lentilactobacillus* (11.1%) representatives dominated the microbial community of this beverage (Figure 6 and Appendix A). The microbial composition of aarts was akin to the khurunga microbial diversity, which might be accounted for by the fact that aarts is a by-product of distilled milk vodka from fermented khurunga (see below). This also explained the high acidity of this sample and the large share of acetic acid bacteria (AAB) in the aarts microbiome, which was the highest among all the studied samples.

***Khurunga*** is a traditional Buryats dairy product, similar to koumiss (see below) but made from cow’s milk. To make khurunga, a special starter (called “ekhe”) is used, which the residents of Buryatia sometimes keep for six months or more [3]. Fermented khurunga is used in the production of aarts, bozo and “togoonay”. Two khurunga samples, 18KR and 26KR, possessed slightly different microbiomes (Figure 6 and Appendix A): *Lactobacillus* representatives (62.4% for 18KR and 94.1% for 26KR) as well as *Lentilactobacillus* spp. (about 4%) were detected in both samples. On the other hand, the microbial community of 18KR also contained 26.8% *Lactococcus*, 3.2% *Streptococcus* and 1.1% *Leuconostoc*, while 26KR had 1.7% *Acetobacter*.

***Koumiss*,** also called “airag”, is a low-alcohol, sour-tasting product made usually from fermented mare’s milk (or camel’s milk), a very popular beverage traditionally produced in Mongolia, Kazakhstan, Kyrgyzstan, and some Central Asian regions of Russia. It is produced by a back-slopping method with raw milk. Fermentation is performed by indigenous LAB and alcoholic fermentation is carried out by yeast [36]. As the national product of nomads, for a long time koumiss was prepared in special leather bags by shaking continuously during horseback riding. As a consequence, stirring for several hours is an essential part of its preparation, favoring the aerobic respiration of the yeasts. 

The bacterial community of the koumiss sample from Buryatia was dominated by *Lactobacillus* (69.7%), *Acetobacter* (18.3%), *Lentilactobacillus* (3%), *Streptococcus* (2.7%) and *Lactococcus* (1.2%) genera (Figure 6 and Appendix A). The microbial communities of the same fermented beverages in the neighboring countries Mongolia, China and Kazakhstan also consisted mainly of *Lactobacillus* representatives [9,21,36,37,38], but the diversity of minor bacterial components was much higher: *Lactococcus*, *Streptococcus*, *Enterococcus*, *Leuconostoc*, *Gluconoacetobacter* and *Acetobacter* were identified in koumisses from these regions. Watanabe and coauthors [39] isolated several species of *Bifidobacterium* from Mongolian airag. 

***Shubat*** is a traditional FMP from Kazakhstan (also named “chal” in Turkmen cuisine), produced from camel’s milk by lactic acid and yeast fermentation [3]. Information on the bacterial composition of this beverage is scarce. To our knowledge, there are only two studies on the bacterial diversity of shubat [21,40], both showing that *Lactobacillus* was the dominant bacterium, making up about 80% of the communities of Chinese shubat, followed by *Acetobacter* and *Streptococcus* representatives [21]. A different pattern was observed in shubat from Altai (Figure 6 and Appendix A): *Lentilactobacillus* (65.8%) and *Lactobacillus* (27.8%) were dominant and the minor components were unclassified *Bifidobacteraceae* (1.7%), *Acetobacter* (1.1%), *Leuconostoc* (0.9%), *Lactococcus* (0.7%) and *Lactiplantibacillus* (0.6%). It should be noted, however, that this difference might be due to improvements in detection technique, i.e., the 16S rRNA gene primers or reference databases used by Yu and coauthors [21] were incapable of distinguishing the closely related [41] *Lentilactobacillus* and *Lactobacillus*. Nevertheless, the microbiome of this fermented beverage was clearly distinct from most of the other analyzed FMP samples, most probably since shubat is the sole product fermented from camel’s milk among all the FMPs analyzed in our study.

***Ayran*** (“shalap” in Kazakhstan) comes from Tatar-Uzbek cuisine and its roots stem to the time of the rise of the Golden Horde. It is a refreshing drink made from a traditional fermented product “katyk” (also called “chegen”, see below) or “suzma” (concentrated and salted katyk) mixed with cold boiled water, or spring water, or mineral water, with ice cubes added [3]. 

The only bacterial phylum dominated in the sampled ayrans was *Firmicutes* (96.3–100% of total microbial communities, Figure 3, Figure 6 and Appendix A). *Lactobacillus* and *Streptococcus* were the two dominant genera. *Lactobacillus* species solely or together with *Streptococcus* were the most numerous bacteria in Karachay-Cherkessia ayrans (samples 5AR, 3AR; 84.9% and 52.8%, respectively), whereas the *Streptococcus* genus was the most abundant in samples from Stavropol Krai (6AR, 13AR, 15AR), making up 69, 63.3, and 90.8% of the total communities, respectively. Additionally, *Acetobacter* representatives were detected as minor components (3.6%) in sample 15AR.

A single exception was an Altai sample (78AR) characterized by significant numbers of *Proteobacteria* (64.1%), mostly representatives of *Pseudomonadales* and *Enterobacteriales*. *Firmicutes* made up only 35.5% and were represented by *Streptococcus*, *Lactobacillus* and *Lactococcus* (15.2, 9.5 and 10.4%, respectively). This result should not be taken into account, as it appears to reflect contamination due to the product’s poor quality control. 

Since in different places in the Caucasus, Stavropol and Siberian regions katyk or suzma are made in different proportions and mixtures with water, the microbial diversity of ayrans varies from place to place. 

***Kefir*** is an ancestral dairy beverage originating from the Northern Caucasus and one of the most popular fermented milk beverages in Russia. Its microbial community consists mainly of LAB, AAB and yeast, naturally packed into a macrostructure known as “kefir grains” that is used as the starter for the fermentation process, carried out at room temperature. During this symbiosis, LAB initiate fermentation of lactose to lactate, also producing volatile metabolites, exopolysaccharides and proteolytic enzymes causing milk proteolysis, while yeast together with AAB produce CO_2_, alcohol and acetate, resulting in the fizzing and acid taste of the final product [5,42]. Representatives of *Firmicutes* were the major components of the microbiomes in two analyzed kefir samples (99.9% for 07KF and 100% for 25KF). Surprisingly, the communities drastically differed from each other at the genus level (Figure 6 and Appendix A). *Lactobacillus* (81.7%) and *Streptococcus* (18.3%) were two most abundant genera found in 07KF, sampled in Stavropol region, while *Lactobacillus* was absent in the sample 25KF from Primorsky region, where the dominant genus was *Lactococcus* (95.8%).

Both *Lactobacillus* and *Lactococcus* species are the most abundant LAB in kefir microbiomes analyzed so far. Common minor components also detected during our work were other LABs—*Streptococcus* and *Leuconostoc* species (Figure 6 and Appendix A). Interestingly, AAB species, which are considered to be another key microorganisms for kefir fermentation, and are dominant in kefirs in many geographical regions [8,42,43,44], were absent in our samples. Moreover, the kefirs from our study had significantly reduced total diversity of lactic acid *Firmicutes* and no bifidobacteria, which might influence the texture and aroma of the product. As a confirmation, the VFA spectrum in kefir samples was limited to lactate and a small concentration of acetate (Figure 2). This is not the first observation that AAB was not detected by NGS in kefir samples: kefirs from Ireland, Belgium and South Africa [45,46] had the same pattern.

***Prostokvasha*** is a traditional FMP in most regions of Russia, Mongolia, Kazakhstan, Kyrgyzstan, Belarus and countries of the Caucasus region. It is made by a back-slopping method with raw milk, and its microbial communities often vary in different regions due to many factors as national traditions, climate, altitude, cow nutrition and habitats. The fraction of *Firmicutes* varied from 61.6% (17PS) to 99.9% (66PS, 69PS, 74PS); in several samples *Proteobacteria* representatives were detected at significant levels (up to 38.4% in 17PS). Roughly all screened prostokvasha could be divided into two groups (Figure 6 and Appendix A): (i) consisted of samples dominated by *Lactococcus* (16PS from Arkhangelsk, 17PS from Tula region, and 29PS, 37PS, 46PS from Dagestan Republic); (ii) contained samples in which *Streptococcus*/*Lactobacillus* consortia or *Streptococcus* domination was observed (Dagestan samples 66PS, 69PS and 74PS). Surprisingly different prostokvasha from the same region—the Dagestan Republic—were found in both groups. It is possible that the differences are due to physical factors, since the products from the first group were from villages located in mountains, while the second one was from villages located in the foothills, closer to the coast of the Caspian Sea. Besides *Lactococcus*, the first group of prostokvasha products contained *Leuconostoc*, unclassified *Lactobacillales* and *Enterococcus* species as minor components. 

The geographical pattern of microbial composition found in the Dagestan samples is in accordance with the results of an analysis of microbiomes of prostokvasha from the neighboring region, Kalmykia republic, also located on the plain. All prostokvasha from Kalmykia were dominated by *Lactobacillus* while the representation of *Lactococcus* was more variable. Beyond that, the Kalmykia prostokvasha differed due to the presence of AAB [19] or *Bifidobacterium* spp. [47], absent in our samples. 

***Ryazhenka*** is a very popular dairy beverage, widely represented in Russia. The main feature of ryazhenka is melted milk used for the fermentation. Thereby it has a buttery, sour-sweet flavor. Standard commercial ryazhenka contains only *Lactobacillus bulgaricus* and *Streptococcus thermophilus* [48] or even only *Streptococcus* representatives [16]. In the ryazhenka sampled in our work, the microbial diversity was much higher (Figure 6 and Appendix A): *Streptococcus* (60.8–98%), *Lactobacillus* (39.2% in 13RZ), *Leuconostoc* (8.1% in 25RZ) and *Lactococcus* (3.8% in 25RZ) species dominated there. From one perspective, this may indicate inconsistencies in production quality control of these farm–made ryazhenka; from another, it may indicate the presence of products with new properties. VFA analysis showed (Figure 2) that in the ryazhenka samples the concentration of lactic acid was the highest among the all analyzed samples of beverages. This fact is in accordance with the domination of homolactic *Streptococcus* spp. yet contradicts with other samples where *Streptococcus* spp. were also dominant (matsoni-like, tan, some ayrans) but the concentration of lactic acid was not so high. 

The homeland of ***tan*** is Armenia, and it is a very popular water-diluted sour-milk refreshment beverage in all the Caucasus. Tan and ayran (see above) are often prepared using similar technology, but unlike ayran, which is made with fresh water, tan is based on a salt solution. Still, the dominant microorganisms in these two dairy products were the same: lactic acid bacteria of the genera *Streptococcus* (54.3%) and *Lactobacillus* (24.5%). Other LAB detected in the tan sample were *Loigolactobacillus* (4.2%), *Leuconostoc* (1.4%), *Lentilactobacillus* (1.3%) and *Lactiplantibacillus* (1.1%) (Figure 6 and Appendix A). A rather significant part of the microbial community (12.1%) was represented by *Proteobacteria* belong to an unknown genus of the order *Enterobacterales*. The differences in the communities are most probably due to the raw materials, especially starters, used to prepare these beverages. 

#### 3.2.2. Creamy Fermented Milk Products

***Matsoni*** (“matzun” in Armenia; “katyk” in Uzbekistan, Tatarstan, Bashkiria, Azerbaydzhan; “yogurt” in Turkmenistan) is a traditional Caucasian sour milk product, and also is very popular in most of the Turkish-speaking countries. The milk for matsoni is preheated (at 90 °C) over a low flame with stirring or in a clay pot in the oven without boiling. Fermentation is started when the milk has cooled to 40 °C [3]. The microbiomes of matsoni-like analyzed products, 5MC and 13MC, were composed almost entirely of *Firmicutes* members (99.9% for both samples). *Streptococcus* was the single dominant genus (97.7%) in 5MC sampled in Karachay-Cherkessia while *Lactobacillus* (1.6%) and *Lactococcus* (0.6%) genera were minor components in the sample (Figure 6 and Appendix A). The sample from Stavropol Krai (13MC) contained 74.3% *Streptococcus* and 25.6% *Lactobacillus*. This is similar to the microbial community of the home-made matsoni from Georgia [49]. The prevalence of the genus *Streptococcus* was the main difference between the matsoni–like product and a similar product’s (chegen, see below) communities and this is explained by the increased temperature during its preparation.

***Chegen*** (“chigee” in China, “tarak” or “tarag” in Buryatia and Mongolia) is the main fermented dairy product of Mongolian cooking [36]. It is usually made of a mixture of milk of different origins, with fermentation conducted at 23–25 °C in a hermetically sealed container [3]. The dominant phylum in chegen samples was *Firmicutes*, representatives of the *Lactobacillales* order in particular, accounting for 97.3–98.7% of all prokaryotes (Figure 6 and Appendix A). *Lactobacillus* species were predominant in both samples (61.1–89.1%), *Lentilactobacillus* made up 37.6% in the 75CG sample and only 4.9% in 76CG. *Acetobacter*, *Lactococcus* and *Streptococcus* strains made up 1.6, 2.4 and 0.5% of the community of the 76CG sample, respectively. *Acetobacter* spp. were also detected as a minor component (1.1%) of the 75CG microbiome. The domination of *Lactobacillus* species in microbial communities in Russian samples correlated well with other studies of the same products in Mongolia and China [9,36,44].

***Sour cream*** (or “smetana”) is a traditional Slavic dairy food. It is produced by manual stirring of cream followed by fermentation using the back-slopping method. The final maturing of sour cream occurs when cold, resulting in specific taste and density. The level of fat varies from 10 to 30%. *Lactococcus* (45.3–88.8%) was dominant in studied sour cream samples with the exception of 22SM, which contained only 16.7% *Lactococcus* (Figure 6 and Appendix A). Other components varied between the samples: *Streptococcus* (0–27.6%), *Leuconostoc* (0–36.3%), *Enterococcus* (6.5% for 20SN), and *Lacticaseibacillus* (14% for 22SM). Similar results showing high bacterial diversity were obtained during studies of sour creams from Buryatia and Kalmykiya by Yu and coauthors in different years [21,47,50]. Significant amounts of proteobacteria belonging to *Klebsiella* (2–32.1%) were detected in Tula (20SN), and in two Stavropol (06SN and 07SN) and one Buryats (20SM) samples. *Acinetobacter* (0.6–7.4%) was found in all sour creams with the exception of 22SM. The presence of these bacteria most probably indicates low quality control during the preparation of the products. 

***Cottage cheese***, or “tvorog”, is a very popular non-liquid dairy product in Russia, obtained by fermenting milk with the subsequent removal of whey. In English-speaking culture, cottage cheese is considered to be a type of young soft cheese, while in the modern Russian-speaking environment cottage cheese is usually not considered a type of cheese. In this work, the tvorog samples had extremely diverse microbiomes composition without clear correlation with the geography of their origin (Figure 6 and Appendix A). In four samples (03TG, 05–07TG), *Streptococcus* species were predominant, accounting from 44.5 to 95.8% of the communities. In the remaining six samples, the genus *Lactococcus* was foundational in the communities (from 59.8 to 86.9%). The main reason for such division was most likely the use of raw (where *Lactococcus* dominated) or pasteurized (where *Streptococcus* dominated) milk for fermentation. Samples 06TG and 20TG (Stavropol and Tula regions) showed significant amounts of *Lactobacillus* (41.2 and 9.5%, respectively). The genus *Leuconostoc* had a significant share only in samples 16TG (Arkhangelsk region) and 17TG (Tula region). *Lactococcus*, followed by *Streptococcus* representatives, dominated in Buryats cottage cheese in the previous study, but also a significant portion of “unwanted” *Proteobacteria* were detected there [20]. Microbiomes of cottage cheese from the Mongolian region were dominated by *Enterococcus*, *Lactococcus*, *Streptococcus* and *Acetobacter* species [21], which differed significantly from our samples from the European part of Russia. 

#### 3.2.3. Cheeses

***Bryndza*** is a type of a brined cheese, traditionally made in Eastern Europe from ewes’ or goat’s milk since ancient times. Currently, cow’s and buffalo’s milk are also used for its preparation. The starting material is renin-precipitated milk-lump cheese, which is subjected to a short fermentation at room temperature followed by grounding and mixing with a salt solution [51]. In our study the dominant genera occurring in bryndza obtained from different regions were *Lactococcus* (48.7–78.5%) and *Streptococcus* (7.6–34.7%). *Leuconostoc* was present as either one of the major (10.9–14.9% for 07BZ, 14BZ and 15BZ, respectively) or minor (1.5 and 2.8% for 03BZ and 05BZ, respectively) components of the microbiomes (Figure 6 and Appendix A). In some samples (03BZ, 07BZ, 14BZ), *Enterococcus* representatives were found in significant amounts (from 1.2% in 14BZ to 5.4% in 07BZ). The proportion of *Proteobacteria* did not exceed 5.1%. Similar results were obtained for Slovakian bryndza, which is a national and a very popular dairy product there and which has microbiomes dominated by *Lactococcus*, *Streptococcus*, *Pediococcus*, and *Enterococcus* members [52]. Additionally, *Lactobacillus* and *Leuconostoc* species were successfully isolated from these cheeses [51]. Although Slovakian bryndza is usually made from ewes’ milk and the Russian products we analyzed were made from cow’s milk, the microbial diversity of these products was highly similar.

***Khuruud*** is a traditional Buryats artisanal cheese, prepared in the manner of other white cheeses by coagulation of casein with rennet extract followed by microbial fermentation, where LAB are the main actors. The formed cheese is placed in a saline solution for maturation. The time (and degree) of maturation determines the texture of the cheese. The microbiome of the khuruud consisted entirely of *Firmicutes* bacteria (Figure 6 and Appendix A): *Lactococcus* (80.4%) and *Leuconostoc* (18.3%). Such diversity is typical for young and soft white-brined cheeses made in Balkan countries [17]. 

Non-spicy cheese ***suluguni*** with a sour milk flavor originates from western Georgia [3]. Unlike other cheeses, after dense casein layers are formed, suluguni is subjected to melting at 80–90 °C and maturation is carried out at 8–12 °C up to two days. More than half of the suluguni-like cheeses’ microbiomes, analyzed here, consisted of lactic acid bacteria belonging to *Firmicutes* (Figure 3). Communities of suluguni-like cheeses sampled in Karachay-Cherkessia differed from each other: 03SU possessed 44.1% *Lactobacillus*, 31.8% *Streptococcus*, 7.2% *Lactococcus*, 6.3% *Leuconostoc*, 4.4% *Latilactobacillus*, while 05SU had 27.2% *Streptococcus*, 17.6% *Lactobacillus*, 15.4% *Leuconostoc*, 12.8% *Enterococcus*, 6.8% *Lactococcus* and 4.1% *Latilactobacillus*. The *Lactococcus* genus (55%) was dominant in 6SU; *Leuconostoc* and *Lacticaseibacillus* were the minor (4.1 and 1.8%, respectively) components of the community (Figure 6 and Appendix A). This heterogeneity is probably due to peculiarities in cheese maturation at each farm. It has been previously shown that the LAB microbiota diversity changes drastically during maturation of cheeses, and that *Lactococcus* and *Leuconostoc* representatives prevail in young cheeses, while *Lactobacillus*, *Enterococcus* and *Streptococcus* species prevail in more matured cheeses due to their lower sensitivity to low pH environments and high salt concentrations [7,17].

## 4. Conclusions

In recent decades, fermented milk products (FMPs) have been intensively studied due to a better understanding of their beneficial properties, such as improved digestion and bioavailability of milk constituents, inhibiting harmful gastrointestinal bacteria, alleviating lactose intolerance and effects on brain activity [15,53,54,55,56]. Lactic acid bacteria, the main actors responsible for milk fermentation, produce a wide range of bioactive compounds that enhance the value of the dairy products, as well as their taste and aroma. Moreover, they produce a variety of extracellular and capsular polysaccharides that contribute to the characteristic textural properties of different types of FMPs [5,18]. 

Using NGS amplicon sequencing, we investigated microbiomes of a variety of home-made FMPs from widely-known kefir, prostokvasha, ryazhenka, koumiss, cottage cheese, sour cream, matsoni-like products and different types of cheeses, to those known only in certain areas’ products, such as khurunga, aarts, chegen, shubat, tan and khuruud. These products were sampled from different regions of Russia, including the Caucasus and Buryatia, known for their ancient and indigenous traditions of FMP production. The microbial communities of some of national Buryats dairy products, such as aarts, khurunga, khuruud, together with tan, ayran and suluguni-like cheese, to our knowledge, were studied here for the first time. Beta-diversity analysis revealed that the majority of samples are forming two clusters. Some products (matsoni-like, ryazhenka, tan, bryndza, khuruud, sour cream, suluguni-like cheese) were only found in one of the clusters, while others (prostokvasha, cottage cheese, kefir) were found in both clusters. *Firmicutes* and *Proteobacteria* were two most numerous phyla in all products, with an overwhelming predominance of *Firmicutes*. This phylum was represented mostly by *Lactococcus*, *Lactobacillus* and *Streptococcus*, followed by *Lentilactobacillus* and *Leuconostoc* species (Figure 6). The main community-forming genus in most of the dairy products made from melted or pasteurized milk or fermented at elevated temperatures (such as ryazhenka, cottage cheese and matsoni-like) was *Streptococcus,* active between 35–40 °C. If raw milk was used in the preparation and the fermentation process took place at room temperature or lower, either *Lactobacillus* (mainly in fermented beverages) or *Lactococcus* (mainly in non-liquid products) members dominated the communities.

In general, based on our results and previous studies, it can be stated that the microbiomes of the same dairy products from different regions were similar in dominant microorganisms and varied mainly in the minor parts of the community. Moreover, different products also might possess quite similar microbial communities, as was observed in our rayzhenka, tan, ayran, matsoni-like, some prostokvasha and cottage cheese samples. This means there is a certain group of basic parameters (temperature of fermentation and other processing parameters, salinity, degree of ripening, etc.) that defines the microbial community of a product regardless of its origin or source of raw milk. On the other hand, the opposite is also true: one type of products may have quite different microbiomes (e.g., sulugini-like cheese, kefir, cottage cheese, prostokvasha), which reflects the source of milk and variation in production routines. 

## Figures and Tables

**Figure 1 microorganisms-10-02140-f001:**
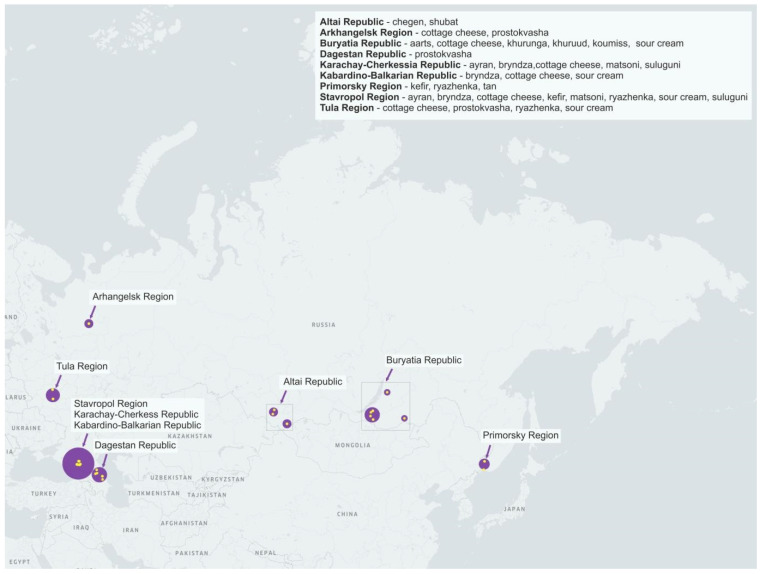
Geographic location of sampling sites.

**Figure 2 microorganisms-10-02140-f002:**
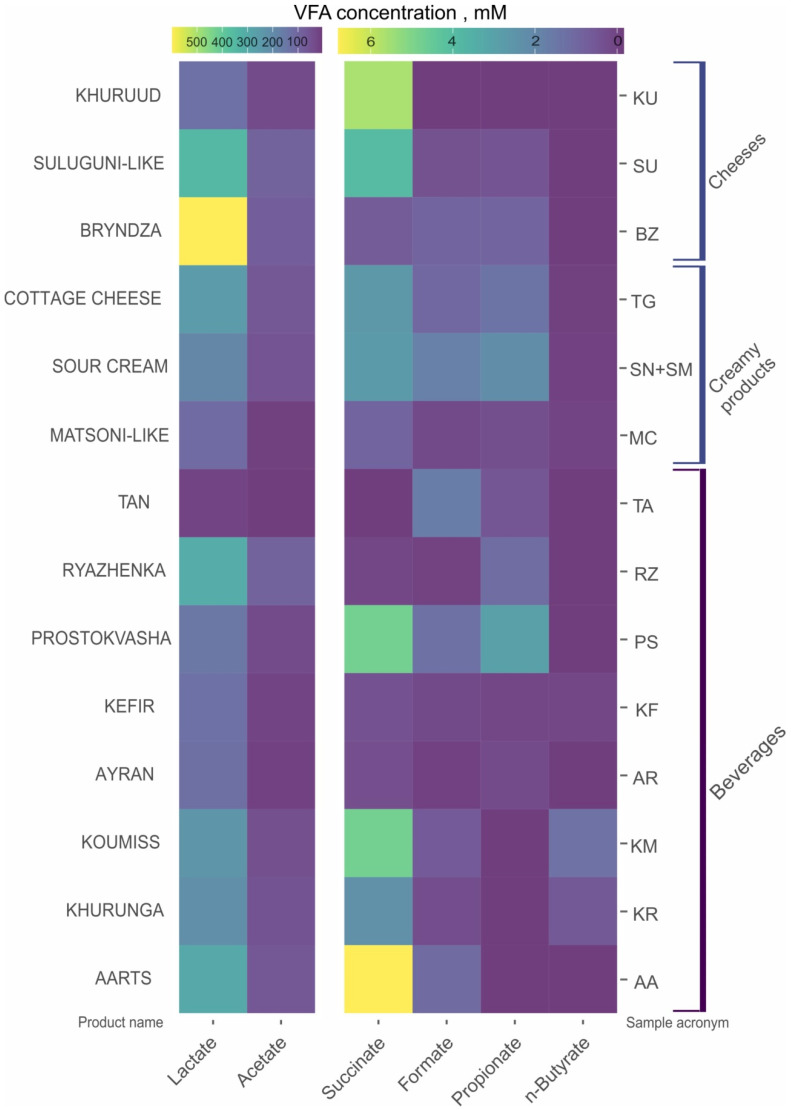
The volatile fatty acids (VFA) content in fermented milk products. Samples from Altai Republic (78AR, 75CG, 76CG and 77SB) were not analyzed.

**Figure 3 microorganisms-10-02140-f003:**
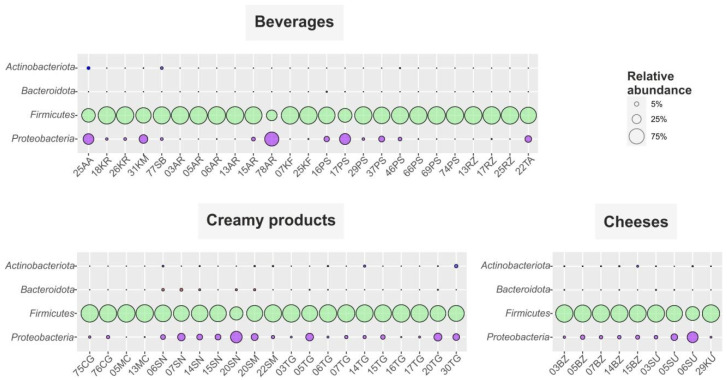
Distribution of FMPs microbial communities on the phylum level. Only phyla containing genera, which made up ≥0.5% of total microbial community for at least one sample, are shown. AA—aarts, AR—ayran, BZ—bryndza, CG—chegen, KF—kefir, KM—koumiss, KR—khurunga, KU—khuruud, MC—matsoni-like product, PS—prostokvasha, RZ—ryazhenka, SB—shubat, SN+SM—sour cream, SU—suluguni-like cheese, TA—tan and TG-cottage cheese.

**Figure 4 microorganisms-10-02140-f004:**
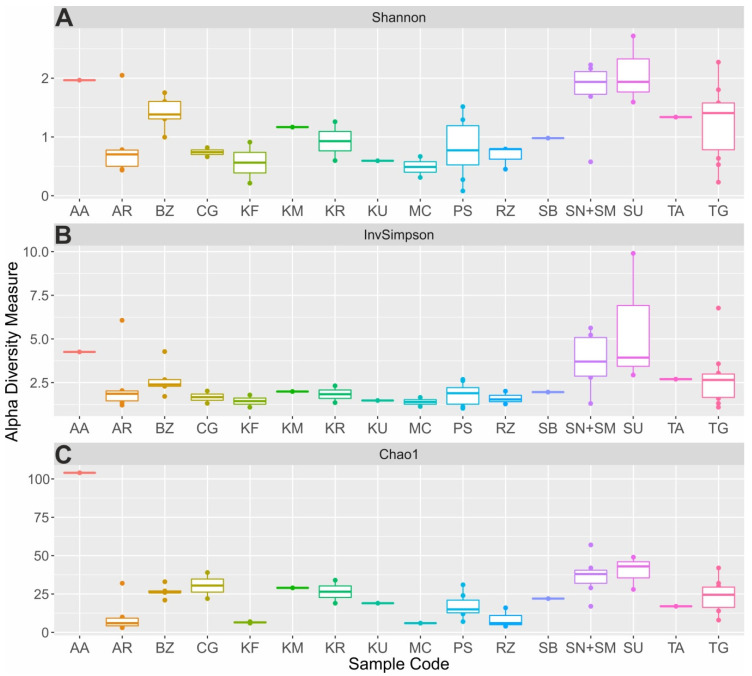
Alpha-diversity metrics of fermented milk products. Calculated values of the Shannon (**A**), inverse Simpson (**B**) diversity indexes and Chao1 richness estimator (**C**) are presented for the aarts (AA), ayran (AR), bryndza (BZ), chegen (CG), kefir (KF), koumiss (KM), khurunga (KR), khuruud (KU), matsoni-like product (MC), prostokvasha (PS), ryazhenka (RZ), shubat (SB), sour cream (SN+SM), suluguni-like cheese (SU), tan (TA) and cottage cheese (TG) samples. *p*-values representing the significance of differences found in alpha-diversity metrics were 0.00681, 0.104 and 2.99e-08, respectively.

**Figure 5 microorganisms-10-02140-f005:**
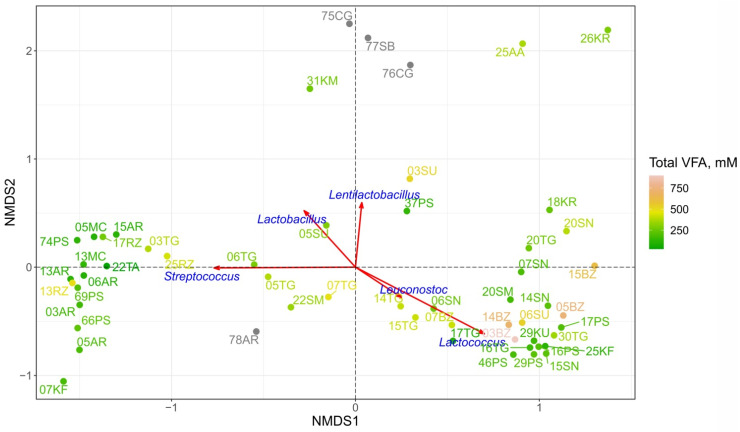
FMP microbiomes dissimilarity displayed by NMDS plots (stress value = 0.13). Color grading corresponds to the concentration of total VFA for each product. The same results for individual VFA are shown on Appendix A. For 4 samples colored in gray, the VFA concentration was not measured. Red vectors indicate the correlation of the dominant genera abundance with the axes of ordination and their statistical significance based on a permutation test (1000 permutations, *p*-value < 0.05), i.e., demonstrate the strength and direction (within the present coordinates) of influence of certain bacterial taxa on sample clustering.

**Figure 6 microorganisms-10-02140-f006:**
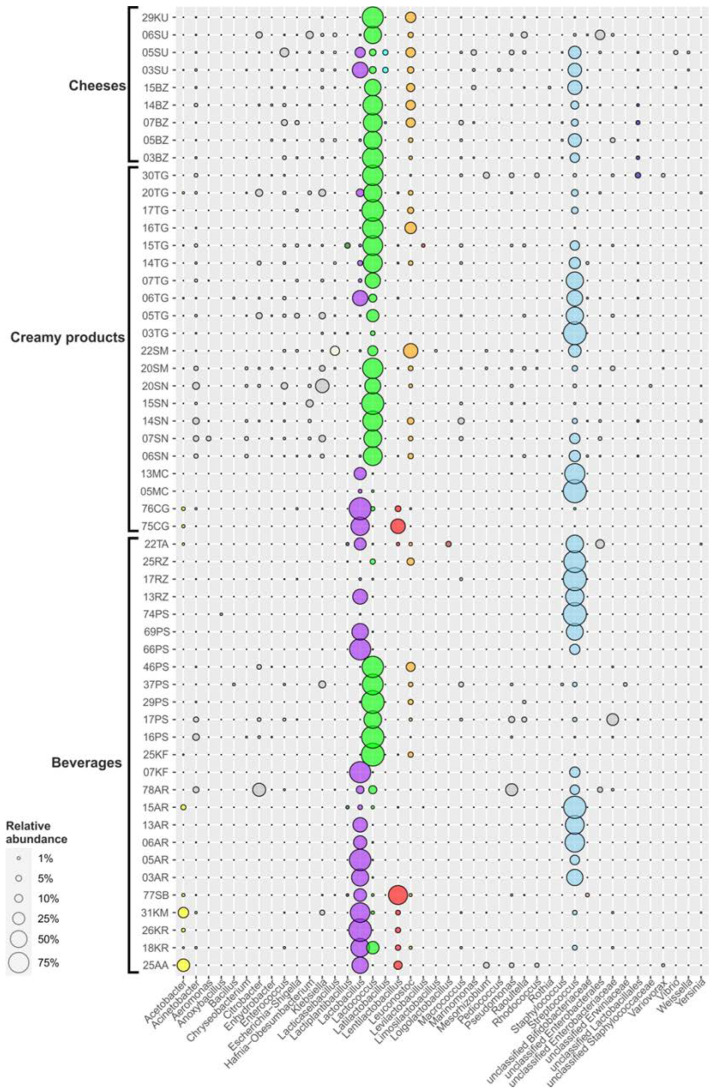
Diversity of bacterial genera found in microbiomes of studied fermented milk products. Only genera which made up ≥0.5% of total microbial community for at least one sample are shown. AA—aarts, AR—ayran, BZ—bryndza, CG—chegen, KF—kefir, KM—koumiss, KR—khurunga, KU—khuruud, MC—matsoni-like product, PS—prostokvasha, RZ—ryazhenka, SB—shubat, SN+SM—sour cream, SU—suluguni-like cheese, TA—tan and TG—cottage cheese.

**Table 1 microorganisms-10-02140-t001:** Samples of fermented milk products, their location and pH.

Sample Type	Sample Acronym	Location	# Of Samples	Sample Designation	pH
Beverages					
Aarts	AA	Buryatia Republic	1	25AA	3.5
Khurunga	KR	Buryatia Republic	2	18KR, 26KR	3.5–4
Koumiss	KM	Buryatia Republic	1	31KM	3.5
Shubat	SB	Altai Republic	1	77SB	ND
Ayran	AR	Karachay-Cherkessia Republic	2	03AR, 05AR	4
		Stavropol Region	3	06AR, 13AR, 15AR	4
		Altai Republic	1	78AR	ND
Kefir	KF	Stavropol Region	1	07KF	4
		Primorsky Region	1	25KF	4.5
Prostokvasha	PS	Arkhangelsk Region	1	16PS	4.5
		Tula Region	1	17PS	4.5
		Dagestan Republic	6	29PS, 37PS, 46PS, 66PS, 69PS, 74PS	3.5–4.5
Ryazhenka	RZ	Stavropol Region	1	13RZ	4.5
		Tula Region	1	17RZ	5
		Primorsky Region	1	25RZ	5
Tan	TA	Primorsky Region	1	22TA	5
Creamy products					
Chegen	CG	Altai Republic	2	75CG, 76CG	ND
Matsoni-like product	MC	Karachay-Cherkessia Republic	1	05MC	4.5
		Stavropol Region	1	13MC	4
Sour cream	SN+SM	Stavropol Region	3	06SN, 07SN, 15SN	5
		Kabardino-Balkarian Republic	1	14SN	4.5
		Tula Region	1	20SN	5
		Buryatia Republic	2	20SM, 22SM	4
Cottage cheese	TG	Karachay-Cherkessia Republic	2	03TG, 05TG	4–5
		Stavropol Region	3	06TG, 07TG, 15TG	4–5.5
		Kabardino-Balkarian Republic	1	14TG	4
		Arkhangelsk Region	1	16TG	5.5
		Tula Region	2	17TG, 20TG	4–5
		Buryatia Republic	1	30TG	4.5
Cheeses					
Bryndza	BZ	Karachay-Cherkessia Republic	2	03BZ, 05BZ	5
		Stavropol Region	2	07BZ, 15BZ	5
		Kabardino-Balkarian Republic	1	14BZ	5
Salt cheese (suluguni-like)	SU	Karachay-Cherkessia Republic	2	03SU, 05SU	5
		Stavropol Region	1	06SU	5.5
Khuruud	KU	Buryatia Republic	1	29KU	6.0

ND—not determined.

## Data Availability

https://www.ncbi.nlm.nih.gov/bioproject/789261 (accessed on 19 September 2022).

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
