# Peer review of "Microbial Communities of Artisanal Fermented Milk Products from Russia"

_microorganisms, 2022, doi:10.3390/microorganisms10112140_

Round 1

Reviewer 1 Report

This publication describes by 16s-V4 methods (not V3-V4 which give more accurate results) the micro-organisms present in 15 traditional products from different regions in Russia. Some of these products have not been analysed before. The main interest is that this publication presents a wide panorama of still poorly described traditional products. The microbiological compositions are complemented by an analysis of several basic products of microbial metabolism. However, this paper remains at a very descriptive level.

In the biodiversity analysis, the authors obtained 367 unique ASVs. However, some ASVs may be very poorly represented or even unique, and it is not mentioned whether the authors filtered out the rarest ASVs and on what basis. It is also not mentioned what the minimum number of reads for each sample is. In general, these analyses start with the rarefaction curve analysis. As a result, the quality of the starting data cannot be assessed.

It is necessary to present in a supplementary table all the results for each sample (in column) of the ASV analysis (in row) indicating the sequence, its taxonomic assignment, the number of reads for each sample, and the overall sum for each ASV.

The ASV analysis stopped at the genus level whereas for many ASVs it is possible to assign species (or at least groups of species). Is that due to the limitation of analysis limited at the V4 region only. This considerably reduces the depth of analysis of the different dairy fermented products.

The authors did not present a principal component analysis of the different products according to ASV. This type of analysis is interesting to show the closeness or difference of the products in a rigorous way. They can thus rely on this analysis in the text as much to show the dispersion within the same product, or the proximity of different products to each other.

Author Response

  • This publication describes by 16s-V4 methods (not V3-V4 which give more accurate results) the microorganisms present in 15 traditional products from different regions in Russia. Some of these products have not been analyzed before. The main interest is that this publication presents a wide panorama of still poorly described traditional products. The microbiological compositions are complemented by an analysis of several basic products of microbial metabolism. However, this paper remains at a very descriptive level.

Indeed, the goal of our study was a description of microbiomes of artisanal fermented milk products including that which were never studied before. This is a first “bird-eye” view which we suppose is being of high importance since the number of these unstudied products is high, their variety is broad and their descriptive analysis will be a great basis for future more in-depth researches including metagenomic/metabolomic as well as cultural or biochemical studies.

Since we have analyzed novel products we suggested that the “coverage” is more important than the “analysis depth”. We used 16S-V4 primers (instead of V3-V4) because these current primers have one of the highest coverage among the 16S rRNA gene fragments universal primers (Sinclair et al., 2015; Zhang et al., 2018; Liu et al., 2020 ). For example there are significant biases with amplification of V3-V4 region of  many novel uncultivated lineages due to variations in length of the V3 region (Vargas-Albores et al., 2017).

Sinclair, O.A. Osman, S. Bertilsson, A. Eiler. Microbial community composition and diversity via 16S rRNA gene amplicons: evaluating the illumina platform. PLoS One, 10 (2) (2015), Article e0116955, 10.1371/journal.pone.0116955

Zhang J, Ding X, Guan R, Zhu C, Xu C, Zhu B, et al. Evaluation of different 16S rRNA gene V regions for exploring bacterial diversity in a eutrophic freshwater lake. Sci Total Environ. 2018;618:1254–67.

Liu et al., 2020, Evaluation of Compatibility of 16S rRNA V3V4 and V4 Amplicon Libraries for Clinical Microbiome Profiling, bioRxiv, doi: 10.1101/2020.08.18.256818

Vargas-Albores F, Ortiz-Suárez LE, Villalpando-Canchola E, Martínez-Porchas M. Size-variable zone in V3 region of 16S rRNA. RNA Biol. 2017 2;14(11):1514-1521. doi: 10.1080/15476286.2017.1317912.

  • In the biodiversity analysis, the authors obtained 367 unique ASVs. However, some ASVs may be very poorly represented or even unique, and it is not mentioned whether the authors filtered out the rarest ASVs and on what basis. It is also not mentioned what the minimum number of reads for each sample is. In general, these analyses start with the rarefaction curve analysis. As a result, the quality of the starting data cannot be assessed.

We did not filter out any ASVs excluding chimeric sequences. On the other hand, on both Figures 3 and 5 only the taxa (phyla or genera, respectively), represented ≥0.5% of total microbial community in at least in one sample were shown which is implying the unique ASVs or poorly represented ASVs are not included. We added a Supplementary table S3 to the revised version, which contains information about the number of reads. Rarefaction curves were also added (Figures S1).

  • It is necessary to present in a supplementary table all the results for each sample (in column) of the ASV analysis (in row) indicating the sequence, its taxonomic assignment, the number of reads for each sample, and the overall sum for each ASV.

Done - Supplementary Table S3.

  • The ASV analysis stopped at the genus level whereas for many ASVs it is possible to assign species (or at least groups of species). Is that due to the limitation of analysis limited at the V4 region only. This considerably reduces the depth of analysis of the different dairy fermented products.

Assigning species based on V4 results leads to a high number of wrong predictions thus we avoid this in our analysis.

  • The authors did not present a principal component analysis of the different products according to ASV. This type of analysis is interesting to show the closeness or difference of the products in a rigorous way. They can thus rely on this analysis in the text as much to show the dispersion within the same product, or the proximity of different products to each other.

We performed NMDS ordination using Bray-Curtis dissimilarity distances and added the results to the revised version (Figures 5 and S2).

Principal component analysis is based on the construction of a correlation matrix of object-attribute relations, when considering metagenomic profiles as an attribute. For our data (predominantly null values and complicated distribution of positive values) PCA is not completely suitable. The preferred NMDS is constructed on a dissimilarity measure that would give the most accurate representation of the relationships of different products according to ASV abundance.

Ramette A. Multivariate analyses in microbial ecology. FEMS Microbiol Ecol. 2007 Nov;62(2):142-60. doi: 10.1111/j.1574-6941.2007.00375.x.

Reviewer 2 Report

Main marks suggested for improvement and clarification of the results provided:

“Fermented milk products prepared by the traditional methods were sampled in local markets in villages and towns of various regions of Russia in 2021 and 2022 during the autumn, spring and summer seasons. For DNA fixation 2 ml aliquots of dairy products were mixed with 2 ml of fixing buffer (100 мМ EDTA, 100 мМ Tris-HCl, 150 мМ NaCl; pH 8.2) at sampling locations. Then the samples were transported to the laboratory at 4°C. DNA extraction and all other manipulations were carried within 7 days after sampling.”

 Point 1 "2 ml aliquots of dairy products" Which was the sampling procedure for the cheese products as their are not in liquid form?

“A total of 1,484,750 reads with an average length of 250 bp were obtained from sequencing of 55 samples. After filtering, denoising and chimera detection 1,217,790 reads were retained representing 367 unique sequences.”

Point 2: There is no information about the number of reads per sample and the normalization for the analysis process. Which were the settings?

“Fixed dairy product samples were centrifuged at 18000 g for 20 minutes and the pellets were used for DNA extraction, which was performed using DNeasy PowerLyzer Microbial Kit (Qiagen, Germany) according to the manufacturer’s instructions, including bead beating stage using FastPrep-24™ 5G grinder (MP Bio, USA). Amplicon libraries of the V4 region of the 16S rRNA gene were prepared as described previously [22] using a pair of primers 515F [23] (5'-GTGBCAGCMGCCGCGGTAA-3') - Pro-mod-805R [24] (5'-GGACTACHVGGGTWTCTAAT-3'). The libraries were sequenced using MiSeq system (Illumina, California, United States). The libraries were prepared and sequenced in two replicates for each sample. All sequencing data were deposited into the NCBI SRA database under BioProject number PRJNA789261 .”

Point 3: In this accession number, it is written that the analysis was performed in Hiseq system and not in Miseq system. 250bp cannot be obtained from Hiseq system. Also, a main point is that the information at the database include more sequences than the sequences referred to the 55 samples of the present analysis. This means that the authors must give us the proper sequences in order to check the analysis. Under these circumstances their work can not be accepted

2.4. Bioinformatics and statistical analysis

Adapter trimming and demultiplexing were performed as described earlier [25]. The obtained reads were filtered and processed using dada2 package v.1.14.1 [26] (parameters: truncLen=220, maxN=0, maxEE=2, truncQ=2) resulting in identification of the amplicon sequence variants (ASV). Taxonomic assignment of ASV was performed using dada2 package v.1.14.1 with native bayesian classifier [27] and Silva 138.1 database [28]. Biodiversity indexes such as Shannon [29], InvSimpson [30], and Chao1 [31] indexes were calculated using the phyloseq v.1.3 package [32]. Visualization of the results was performed with ggplot2 package (https://ggplot2.tidyverse.org.).

 Point 4:Which were the settings for biodiversity indexes?

A total of 1,484,750 reads with an average length of 250 bp were obtained from sequencing of 55 samples. After filtering, denoising and chimera detection 1,217,790 reads were retained representing 367 unique sequences. The obtained ASV were assigned to more than two hundreds  genera within 32 phyla, but more than 95% of total number of sequences were affiliated to Firmicutes and Proteobacteria (Figure 3)”.

Point 5: In figure 3,  the microbial communities are visualized in phylum level for 17 types of cheeses. In contrast, in figure 4, information for alpha diversity is missing for SM sour cream cheese.

Point 6: Reads or raw reads? Τhere is no information about the average of reads per samples. It would be useful the addition of the rarefraction curves in order to check the run analysis. Should be followed the same nomenclature - taxonomy in the text for phylum level. 

“To estimate overall diversity in all samples analyzed the alpha-diversity indexes were calculated for each sample. According to the Shannon Index, suluguni-like cheeses, sour cream samples and aarts possessed the highest biodiversity in comparison with other studied products (Figure 4A)”

“The analysis of the diversity based on the inverse Simpson and Chao1 indexes revealed similar trends (Figures 4Band 4C). The highest Chao1 index varied from 3 (05AR) to 80 (25AA), while inverse Simpson values – from 1.02 (74PS) to 9.9 (05SU )”.

Point 7: There are no p-values in figure 4 and 5

Point 8: Chao1 is not an index, only Shannon and Inverse Simpson

Point 9: How has been calculated the similarity of this trend between the three indices? There are 3 different scales for the alpha diversity measurement, in figure 4.

Point 10: In figure 3, there is visualized the microbial communities in phylum level for 17 types of cheeses. In contrast, in figure 4, information for alpha diversity is missing for SM sour cream cheese.

Point 11: In Figure 5, how the relative abundances between the group of cheeses have been calculated? Missing information at the figure. It is not clear is there are statistics.

“Apparently, due to the limitations of amplicon sequencing, this method can hardly be used to identify specific details that distinguish different FMP or FMP from different regions and other approaches (as meta-omics) should be implemented”

 Point 12: As there is no beta-diversity analysis, how can be compared the microbiota from different type of fermented foods and conclude that the amplicon sequencing cannot differentiate samples from the regions the samples have been taken. So, the main conclusion should be altered according to the elements provided in the paper and not by hypothetic positions. Without the beta -diversity analysis, there are no p-values among the group samples. 

There must be precise and adequate explanation regarding the sequencing analysis for the acceptance and simultaneously improvement of the results  

Author Response

  • “Fermented milk products prepared by the traditional methods were sampled in local markets in villages and towns of various regions of Russia in 2021 and 2022 during the autumn, spring and summer seasons. For DNA fixation 2 ml aliquots of dairy products were mixed with 2 ml of fixing buffer (100 мМ EDTA, 100 мМ Tris-HCl, 150 мМ NaCl; pH 8.2) at sampling locations. Then the samples were transported to the laboratory at 4°C. DNA extraction and all other manipulations were carried within 7 days after sampling.”

 Point 1 "2 ml aliquots of dairy products" Which was the sampling procedure for the cheese products as their are not in liquid form?

Cheeses were sampled by a 20-ml sterile syringe with its front end cut off.

We have added this to the revised manuscript.  

  • “A total of 1,484,750 reads with an average length of 250 bp were obtained from sequencing of 55 samples. After filtering, denoising and chimera detection 1,217,790 reads were retained representing 367 unique sequences.”

Point 2: There is no information about the number of reads per sample and the normalization for the analysis process. Which were the settings?

We added a Supplementary Table S3 containing this information to the revised version of the manuscript. We used relative abundances (see the answer to Point 11) to analyze taxonomic composition in the studied microbiomes.

  • “Fixed dairy product samples were centrifuged at 18000 g for 20 minutes and the pellets were used for DNA extraction, which was performed using DNeasy PowerLyzer Microbial Kit (Qiagen, Germany) according to the manufacturer’s instructions, including bead beating stage using FastPrep-24™ 5G grinder (MP Bio, USA). Amplicon libraries of the V4 region of the 16S rRNA gene were prepared as described previously [22] using a pair of primers 515F [23] (5'-GTGBCAGCMGCCGCGGTAA-3') - Pro-mod-805R [24] (5'-GGACTACHVGGGTWTCTAAT-3'). The libraries were sequenced using MiSeq system (Illumina, California, United States). The libraries were prepared and sequenced in two replicates for each sample. All sequencing data were deposited into the NCBI SRA database under BioProject number PRJNA789261 .”

Point 3: In this accession number, it is written that the analysis was performed in Hiseq system and not in Miseq system. 250bp cannot be obtained from Hiseq system. Also, a main point is that the information at the database include more sequences than the sequences referred to the 55 samples of the present analysis. This means that the authors must give us the proper sequences in order to check the analysis. Under these circumstances their work can not be accepted

There was a mistake made during the submission to SRA. Indeed we used the Miseq system and SRA metadata was corrected within the PRJNA789261 bioproject, accordingly. We also added a Supplementary Table (Table S1) with SRX accession numbers of the samples analyzed in the study to the revised version of the manuscript.

  • 4. Bioinformatics and statistical analysis

Adapter trimming and demultiplexing were performed as described earlier [25]. The obtained reads were filtered and processed using dada2 package v.1.14.1 [26] (parameters: truncLen=220, maxN=0, maxEE=2, truncQ=2) resulting in identification of the amplicon sequence variants (ASV). Taxonomic assignment of ASV was performed using dada2 package v.1.14.1 with native bayesian classifier [27] and Silva 138.1 database [28]. Biodiversity indexes such as Shannon [29], InvSimpson [30], and Chao1 [31] indexes were calculated using the phyloseq v.1.3 package [32]. Visualization of the results was performed with ggplot2 package (https://ggplot2.tidyverse.org.).

 Point 4:Which were the settings for biodiversity indexes?

Biodiversity indexes were calculated using phyloseq 1.3 with default settings.

  • A total of 1,484,750 reads with an average length of 250 bp were obtained from sequencing of 55 samples. After filtering, denoising and chimera detection 1,217,790 reads were retained representing 367 unique sequences. The obtained ASV were assigned to more than two hundreds genera within 32 phyla, but more than 95% of total number of sequences were affiliated to Firmicutes and Proteobacteria (Figure 3)”.

Point 5: In figure 3,  the microbial communities are visualized in phylum level for 17 types of cheeses. In contrast, in figure 4, information for alpha diversity is missing for SM sour cream cheese.

Both SN and SM samples belonged to the same product type – sour creams (this was indicated in Table 1). We corrected the figures to make it more clear.

  • Point 6: Reads or raw reads? Τhere is no information about the average of reads per samples. It would be useful the addition of the rarefraction curves in order to check the run analysis. Should be followed the same nomenclature - taxonomy in the text for phylum level.

The sentence was rewritten with consideration to the Reviwer`s comments.  A Supplementary Table S3 containing information about the number of reads per sample was added to the revised version. Rarefaction curves were also added (Figures S1).

  • “To estimate overall diversity in all samples analyzed the alpha-diversity indexes were calculated for each sample. According to the Shannon Index, suluguni-like cheeses, sour cream samples and aarts possessed the highest biodiversity in comparison with other studied products (Figure 4A)”

“The analysis of the diversity based on the inverse Simpson and Chao1 indexes revealed similar trends (Figures 4Band 4C). The highest Chao1 index varied from 3 (05AR) to 80 (25AA), while inverse Simpson values – from 1.02 (74PS) to 9.9 (05SU )”.

Point 7: There are no p-values in figure 4 and 5

One-way ANOVA test was performed to calculate the alpha-diversity indexes, obtained P-values were added to legend of Figure 4. We assume that P-values are not necessary for Figure 5 (now it is a Figure 6) since it is a reflection of direct observation, not a result of a test of statistical hypothesis.

  • Point 8: Chao1 is not an index, only Shannon and Inverse Simpson

Corrected.

  • Point 9: How has been calculated the similarity of this trend between the three indices? There are 3 different scales for the alpha diversity measurement, in figure 4.

Indeed we estimated only the general patterns by comparing values of the indexes for different samples. We suppose the box-plots of the Figure 4 are quite informative itself and the trendlines are not necessary and might only add an excessive information noise to this already quite "dense" figure. This sentence was rewritten in the revised version of the manuscript.

  • Point 10: In figure 3, there is visualized the microbial communities in phylum level for 17 types of cheeses. In contrast, in figure 4, information for alpha diversity is missing for SM sour cream cheese.

Both SN and SM samples belonged to the same product type – sour creams (this is indicated in Table 1). In the revised version of the manuscript we corrected the figure to make it more clear.

  • Point 11: In Figure 5, how the relative abundances between the group of cheeses have been calculated? Missing information at the figure. It is not clear is there are statistics.

We did not calculate relative abundances between groups of the fermented milk products but the relative abundance of microbial genera. The relative abundance of microbial genera was calculated for each sample as follows: sum of reads affiliated to ASVs belonged to the genus divided by a total number of reads obtained for the sample. To compare relative abundances of ASVs between the samples we used NMDS ordination (see Fig. 5 in the revised version of manuscript) which was added to the revised version of the manuscript.

  • “Apparently, due to the limitations of amplicon sequencing, this method can hardly be used to identify specific details that distinguish different FMP or FMP from different regions and other approaches (as meta-omics) should be implemented”

 Point 12: As there is no beta-diversity analysis, how can be compared the microbiota from different type of fermented foods and conclude that the amplicon sequencing cannot differentiate samples from the regions the samples have been taken. So, the main conclusion should be altered according to the elements provided in the paper and not by hypothetic positions. Without the beta -diversity analysis, there are no p-values among the group samples.

We agree with the Reviewer that our conclusions were not supported statistically. We analyzed beta-diversity with NMDS ordination using Bray-Curtis dissimilarity and found that the samples formed two main clusters. Some FMP were cluster-specific (i.e. belong only to one of the clusters), others were distributed within both clusters. Thus we still agree with our previous thesis that amplicon sequencing itself cannot distinguish at least some of the product types from the other and to do so additional approaches are needed. However, taking into account the criticism of the Reviewer and that these rather general statements (about other omics) do not carry any additional meaning, we decided to delete this part in the revised version of the manuscript.

Round 2

Reviewer 1 Report

The modifications were carried out correctly. The data are well presented. However; the lack of species discrimination due to the use of a poorly resolving marker considerably reduces the interest of this article. Its interest comes therefore from the presence of a wide range of traditional products.

Reviewer 2 Report

  • Reading the revised paper as well as the supplementary material, I see that the points I have raised have been either corrected as suggested, or the text has now been amended to provide better clarification and understanding. The authors responded to all 12 points of suggestions, having improved all parts of the text, analysing in a better way their methodology and the specific sections presented in a way to reflect the impact of their results. Efforts have been made to explain statistics they followed, figures have been corrected and paragraphs have been modified to address in a better way their results.
  • The paper can be accepted in the present form.     
    .
  •